# Diesel Exhaust Particle Exposure Compromises Alveolar Macrophage Mitochondrial Bioenergetics

**DOI:** 10.3390/ijms20225598

**Published:** 2019-11-09

**Authors:** Jonathan L. Gibbs, Blake W. Dallon, Joshua B. Lewis, Chase M. Walton, Juan A. Arroyo, Paul R. Reynolds, Benjamin T. Bikman

**Affiliations:** 1Metabolism Research Lab, Department of Physiology and Developmental Biology, Brigham Young University, Provo, UT 84602, USA; 2Lung and Placental Research Lab, Department of Physiology and Developmental Biology, Brigham Young University, Provo, UT 84602, USA

**Keywords:** diesel exhaust, inflammation, mitochondria, ceramides

## Abstract

Diesel exhaust particles (DEPs) are known pathogenic pollutants that constitute a significant quantity of air pollution. Given the ubiquitous presence of macrophages throughout the body, including the lungs, as well as their critical role in tissue and organismal metabolic function, we sought to determine the effect of DEP exposure on macrophage mitochondrial function. Following daily DEP exposure in mice, pulmonary macrophages were isolated for mitochondrial analyses, revealing reduced respiration rates and dramatically elevated H_2_O_2_ levels. Serum ceramides and inflammatory cytokines were increased. To determine the degree to which the changes in mitochondrial function in macrophages were not dependent on any cross-cell communication, primary pulmonary murine macrophages were used to replicate the DEP exposure in a cell culture model. We observed similar changes as seen in pulmonary macrophages, namely diminished mitochondrial respiration, but increased H_2_O_2_ production. Interestingly, when treated with myriocin to inhibit ceramide biosynthesis, these DEP-induced mitochondrial changes were mitigated. Altogether, these data suggest that DEP exposure may compromise macrophage mitochondrial and whole-body function via pathologic alterations in macrophage ceramide metabolism.

## 1. Introduction

Animals have two main interfaces where the body directly incorporates and transports material from the external to the internal environments. Increasingly, these nexuses of interaction, the intestines and the lungs, are scrutinized in the quest to fully understand the metabolic ramifications of the immediate environment around us. Despite the explosion in interest regarding intestinal health and metabolic function, particularly the microbiome, the air we breathe, though perhaps made of a subtler substance, nevertheless elicits a substantial effect on metabolic physiology.

Optimal cardiometabolic health requires cooperative input from the immune system. One of the most impactful and recent discoveries in biomedical science was the realization that the metabolic and immune systems, both essential to organismal survival, not only interact, but indeed rely on each other. One of the earliest observations of this interaction was the production of inflammatory cytokines from adipose, a prototypical metabolic tissue, which resulted in defects in systemic sensitivity to insulin, a prototypical metabolic hormone [1]. Macrophages, whether adipose or otherwise, are highly relevant to this process, with a prominent line of evidence suggesting that macrophage activation provides an inflammatory “spark” that ignites systemic downstream cardiometabolic complications [2,3,4]. Insofar as the lung, being one of two sites of direct environment–host interaction, is enriched with macrophages, evidence implicating the macrophage and its inflammatory actions in metabolic disruption in response to inhaled noxious stimuli is strong [5]. 

Inhaled diesel exhaust elicits potent effects on the body, rapidly activating myriad inflammatory processes as the diesel particles (DEPs) affect varying levels of the respiratory tree [6]. Unsurprisingly, consistent DEP inhalation is implicated with several cardiorespiratory disorders and mortality [7], with both epidemiological [8] and molecular [7] evidence indicating a primary role of the lung macrophage. At least some of the pathology associated with noxious stimuli exposure is a result of macrophage-induced cytokine release, which has been a focus of earlier work from our lab [4,9,10].

Interestingly, while mitochondria are an understandably prominent focus in the context of exploring metabolic health, its influence in mediating macrophage-induced metabolic disruption is lacking and the limited evidence overwhelmingly exists in non-lung tissues. We have previously found that cigarette smoke exposure alters whole body metabolic function [4,9,11], including mitochondria-specific consequences [3]. However, while cigarette smoke exposure is declining in many parts of the world, the increasing industrialization is leading to evermore exposure to diesel exhaust combustion [12]. The purpose of this study was to elucidate the effects of diesel exhaust particles, which are readily inhaled through the lungs, on the bioenergetic profile of macrophages.

## 2. Results 

### 2.1. Lung Macrophage Mitochondrial Function Is Altered in Mice Following DEP Exposure

The primary outcome for these experiments was to determine the degree to which diesel exhaust particles (DEPs) disrupt pulmonary macrophage bioenergetics. Following four weeks of DEP exposure, alveolar macrophage mitochondrial respiration was significantly different compared with those from room air controls (CON). Specifically, while complex I-mediated leak respiration (GM*_L_*) was similar between conditions (Figure 1A), the difference in respiration rates became obvious with the addition of ADP (GM*_P_*), suggesting a defect with oxidative phosphorylation. The disparate rates were maintained with the addition of succinate (GMS*_P_*) to scrutinize the contribution of complex II. As a general test of mitochondrial “fitness,” the respiratory control ratio (RCR) revealed a significant reduction in macrophages from DEP-exposed mice (Figure 1B).

In an attempt to understand the use of mitochondrial oxygen, we measured H_2_O_2_ levels from the mouse-derived alveolar macrophages. Even though DEP macrophages utilized less oxygen (stated above), the amount of the reactive oxygen species H_2_O_2_ generated was significantly greater (Figure 1C), which was further reflected in the comparison of H_2_O_2_ as a function of O_2_ use (Figure 1D).

### 2.2. DEP Increases Plasma Cytokines and Ceramides

Upon noting the significant increases in H_2_O_2_, and bearing in mind the connection between oxidative stress and inflammation [13], we sought to establish whether this oxidative stress was mirrored in a systemic shift in the inflammatory profile. Plasma levels of both IL-1β (Figure 2A) and TNF-α (Figure 2B) were measured, revealing a several-fold increase in each.

We have previously shown that heightened inflammatory signaling increases the biosynthesis of ceramides [14], including induction via TNF-α [15]. Plasma ceramides following DEP exposure were significantly increased compared with controls (CON), including increases in multiple specific ceramide species (Figure 3).

### 2.3. DEP Alters Primary Alveolar Macrophage Mitochondrial Function

We next sought to determine the degree to which our observations of altered mitochondrial function with DEP exposure are autonomous to the alveolar macrophage, as well as the necessity of ceramides in this response. Accordingly, we isolated alveolar macrophages from 16-week-old mice and plated for cell culture. These primary macrophages were incubated with DEP (or vehicle) for 12 h, followed by cell harvesting and mitochondrial assessments. Given the increased plasma ceramides in the murine model, we also included the use of myriocin (MYR), a known inhibitor of ceramide biosynthesis in the culture medium (Figure 4). To determine whether ceramides elicit mitochondrial effects distinct from DEP exposure, C2-ceramide was included in combination with DEP, to no effect [16].

Primary alveolar macrophages responded to DEP in a similar manner as before, with reduced O_2_ flux and RCR (Figure 5A,B). Moreover, H_2_O_2_ emission was significantly elevated (Figure 5C,D). However, these changes were mitigated in the myriocin and DEP (DEP+MYR). 

## 3. Discussion

Globally, an interesting correlational trend is becoming obvious—countries that are experiencing the greatest degree of industrial development are experiencing parallel changes in declining metabolic health [17,18]. Of the myriad pollutants surrounding us, diesel exhaust particles (DEP) are known noxious stimuli that harm the body, including metabolic function. Mitochondria are both at the nexus of metabolic health and countless pathological processes, including insulin resistance and type 2 diabetes, with multiple lines of evidence implicating a role for mitochondria-induced oxidative stress [19] and inflammation [20] as necessary mediators. Considering the astounding and growing prevalence of metabolic diseases [21,22,23], the purpose of these studies was to identify the degree to which an increasingly common environmental pollutant, diesel exhaust, can disrupt normal mitochondrial function and exacerbate metabolic physiology. Our evidence reveals that diesel exhaust particles (DEPs) are capable of significantly altering macrophage bioenergetics in the lung, likely via ceramide accrual, along with systemic adaptations including increased inflammation. 

Given not only their location in the lung, but also their ability to respond to noxious stimuli, macrophages are important cellular “gate keepers” at the interface between environment and organism. Ample evidence has established that DEP drives several immunomodulatory changes in macrophages, resulting in changes not only in inflammatory status [24] but also oxidative stress [25,26]. Our initial observations to identify the mitochondrial effect of DEP revealed a significant reduction in oxygen use yet a concomitant increase in H_2_O_2_ production. Moreover, the respiratory control ratio, a function of oxidative phosphorylation responsiveness, was blunted with DEP treatment. Corroborating previous work [27], this overall shift in mitochondrial phenotype and the generation of oxidative stress could be responsible for the dramatically increased systemic inflammatory cytokines. 

Alveolar macrophages exist in an oxygen-rich milieu. We submit that the H_2_O_2_:O_2_ ratio is a valuable outcome in identifying the use of oxygen in alveolar macrophages—namely, whether oxygen use in the mitochondria is used for the production of energetic (e.g., ATP) or signaling (e.g., H_2_O_2_) molecules. Importantly, oxidative stress molecules (i.e., O_2_^−^, H_2_O_2_, etc.) induce the expression of inflammatory signals in macrophages [28]. Future efforts will reveal the degree to which reactive oxygen species are necessary in DEP-induced macrophage cytokine production. 

Although the precise process of events remains unknown, these results implicate ceramides as necessary in the DEP-induced mitochondrial changes. Our findings of a possible role for ceramides as a mediator of DEP-induced macrophage changes present several possible future directions. In particular, diesel exhaust particles (DEPs) are known to activate inflammatory pathways relevant to insulin resistance, such as the receptor for advanced glycation end-products (RAGE) [29] and toll-like receptor (TLR) 4 [14], both of which we have previously explored. Work in the immediate future will reveal the relevance of these receptors in mediating DEP-induced macrophage mitochondrial changes. 

These results provide mechanistic insight into the long-observed association between air pollution and cardiometabolic disruption [17,18]. That ceramides may be necessary in conducting the mitochondrial and systemic damage from DEP introduces the provocative and tentative conclusion that anti-ceramide therapies could partially offset some of the consequences of air pollution containing DEP. Ceramides impact numerous aspects of heart disease, including cardiomyopathy and hypertension [30,31], extending the relevance of these findings. 

## 4. Materials and Methods

### 4.1. Animals

Male C57Bl/6 mice were housed in a conventional animal vivarium and maintained on a 12:12 h light–dark cycle. Animals received a standard diet chow (Harlan-Teklad 8604) and water ad libitum. At 16 weeks of age, animals were randomly divided into room air- and diesel exhaust particle (DEP)-exposed groups for four weeks. Mice were placed in soft restraints and connected to the exposure tower of a nose-only exposure system (InExpose System; Scireq, Montreal, QC, Canada). Animals were nasally exposed to mainstream DEP generated as outlined below. Tissues were harvested at the conclusion of the study period and following a 6 h fast and 24 h after the final exposure. Lungs were lavaged for assessment and capturing of bronchoalveolar lavage fluid (BALF), as described previously [11,32]. Total numbers of cells were counted using a hemocytometer with trypan blue exclusion. Cells were either immediately utilized for further analysis (e.g., lipid extraction and mitochondrial respiration) or plated on cell culture dishes for further treatment. Studies were conducted in accordance with the principles and procedures outlined in the National Institutes of Health *Guide for the Care and Use of Laboratory Animals* and were approved by the Institutional Animal Care and Use Committee (IACUC) at Brigham Young University (18-0101; 26 January 2018).

### 4.2. Diesel Exhaust Particles

The Standard Reference Material (SRM-2975) maintained by the National Institute of Standards & Technology (NIST) was used in the current investigation. This reference DEP was selected due to its wide accessibility and known ability to permit penetration into the alveolar compartment (PMID 24330719). SRM-2975 was originally obtained from M.E. Wright of the Donaldson Company, Inc., Minneapolis, MN, as outlined previously (PMID 21087909). Briefly, the DEP was generated by heavy-duty forklift engines and collected from a filtering system specifically designed for diesel-powered forklifts. The reported mean diameter of course SRM 2975, determined by area distribution light scattering, was 11.2 ± 0.1 μm. However, the mean particle size computed by number distribution was 1.62 ± 0.01 μm. To our knowledge, information regarding load, fuel, and other run conditions are not available. As previously described, DEP was generated so that animals received a nebulized dose of 15 ng of freshly vortexed DEP per exposure in a bolus of approximately 20 μL, which translates to roughly 3 µg/mL in the plasma, considered a physiological dose [7]. Control animals were given the equivalent volume, but instead only received nebulized PBS.

### 4.3. Primary Alveolar Macrophage Culture

To obtain primary alveolar macrophages (AMs), four WT C57Bl6 mice per experiment were sedated and exsanguinated to ensure euthanasia as outlined previously [7]. Bronchoalveolar lavage fluid (BALF) was specifically harvested through the instillation and recovery of seven 1 mL boluses of PBS with a syringe attached to a catheter for a total of 7 mL. Each 7 mL BALF sample was centrifuged at 1000 rpm for 10 min. The cell pellet was resuspended in warm DMEM, pooled from the four mice, and equal concentrations of approximately 50,000 cells were plated and allowed to adhere overnight before exposure. Where indicated, cells were treated singularly or in combination with the following reagents: DEP (3 µg/mL); *N*-acetyl-d-sphingosine (C2-ceramide; 20 µM); myriocin (10 μM, Sigma M1177, St. Louis, MO, United States), a known and widely used inhibitor of ceramide biosynthesis. 

### 4.4. Mitochondrial Respiration

Cells and tissue were prepared for mitochondrial respiration as described previously [9] before being transferred to respirometer chambers using the Oroboros O2K oxygraph (Oroboros, Innsbruck, Austria). Electron flow through complex I was supported by glutamate + malate (10 mM and 2 mM, respectively) to determine leak oxygen consumption (GML). Following stabilization, adenosine diphosphate (ADP) (2.5 mM) was added to determine oxidative phosphorylation capacity (GMD). Succinate was added (GMSD) for complex I + II electron flow into the Q-junction. Lastly, residual oxygen consumption was measured by adding antimycin A (2.5 μM) to block complex III action, effectively stopping any electron flow, which provides a baseline rate of respiration. Respiratory control ratio (RCR) was determined as the ratio of GM*_P_*/GM*_L_*. Following respiration protocol, samples were removed from the chambers and used for further analysis, including protein quantification.

### 4.5. H_2_O_2_ Emission

H_2_O_2_ emission was measured using an Amplex Red Hydrogen Peroxide/Peroxidase Assay kit (Molecular Probes; A22188, Eugene, OR, United States) as described previously [33]. A reaction mixture containing 50 μM Amplex Red and 0.1 unit/mL HRP in KRPG (Krebs-Ringer phosphate glucose, Sigma) buffer was prepared (145 mM NaCl, 5.7 mM sodium phosphate, 4.86 mM KCl, 0.54 mM CaCl2, 1.22 mM MgSO4, and 5.5 mM glucose). The reaction mixture was pre-warmed in a 96-well plate with 100 μL of mixture per well. A 20 μL aliquot of cells suspended in KRPG buffer (~1.5 × 104 cells) were added to each well. Samples were incubated for 1 h. Fluorescence was measured with a microplate reader (Molecular Devices; San Jose, CA, USA).

### 4.6. Tissue Homogenization

As previously described [34], muscles were ground-glass homogenized in lysis buffer (50 mM Tris-HCl, pH 7.4; 250 mM mannitol; 50 mM NaF; 5 mM sodium pyrophosphate; 1 mM EDTA; 1 mM EGTA; 1% Triton X-100; 50 mM β-glycerophosphate; 1 mM sodium orthovanadate; 1 mM DTT; 1 mM benzamidine; 0.1 mM phenylmethane sulfonyl fluoride; 5 μg/mL soybean trypsin inhibitor). This was centrifuged at 10,000× *g* at 4 °C for 10 min. The supernatant was retained for Western blotting analysis.

### 4.7. Ceramide Analysis

For quantification of ceramides from plasma and cell lysate, 100 μL of sample was resuspended in 300 μL ice-cold phosphate-buffered saline (PBS) and 1.5 mL of methanol, as described previously [35]. Samples were centrifuged and supernatant was transferred to a clean tube. Following the addition of 30 μL of 1 M KOH in methanol, samples were incubated overnight at 4°C. Samples were dried to 50% volume, and 25 μL glacial acetic acid was added to neutralize KOH. Separation of aqueous and organic phases required the addition of 300 μL LC-grade chloroform and 600 μL DDH_2_O followed by centrifugation for 2 min at maximum speed. The lower organic phase was transferred to a fresh vial. This separation step was repeated twice. All lipid samples were dried in a vacuum centrifuge (Eppendorf Concentrator Plus, Hamburg, Germany). Lipids were quantified by shotgun lipidomics using an ABI 5600+ (AB Sciex, Framingham, MA, United States). Briefly, we simultaneously identified changes in hundreds of distinct lipid species via a nonbiased approach following direct infusion of extracted lipids containing 18 mM ammonium fluoride to aid in ionization of neutral lipids and to reduce salt adducts. Data from the AB Sciex 5600+ were collected and calibrated with Analyst and PeakView software (AB Sciex). The in-house-developed Lipid Explorer software assists with simplifying the data by identifying lipid species based on exact mass and fragmentation patterns. 

### 4.8. ELISAs 

ELISA kits that specifically screen for mouse TNF-α or IL-1β (Ray BioTech, Norcross, GA, United States) were used to assess secretion of the two inflammatory cytokines. Briefly, serum samples were isolated from control and treatment groups and screened as outlined in the manufacturer’s instructions.

### 4.9. Statistical Methods

Data are presented as means ± SEM. Data were compared with a Student’s *t*-test (Graphpad Prism, San Diego, CA, United States; Microsoft Excel, Redmond, WA, United States). Significance was set at *p* < 0.05.

## Figures and Tables

**Figure 1 ijms-20-05598-f001:**
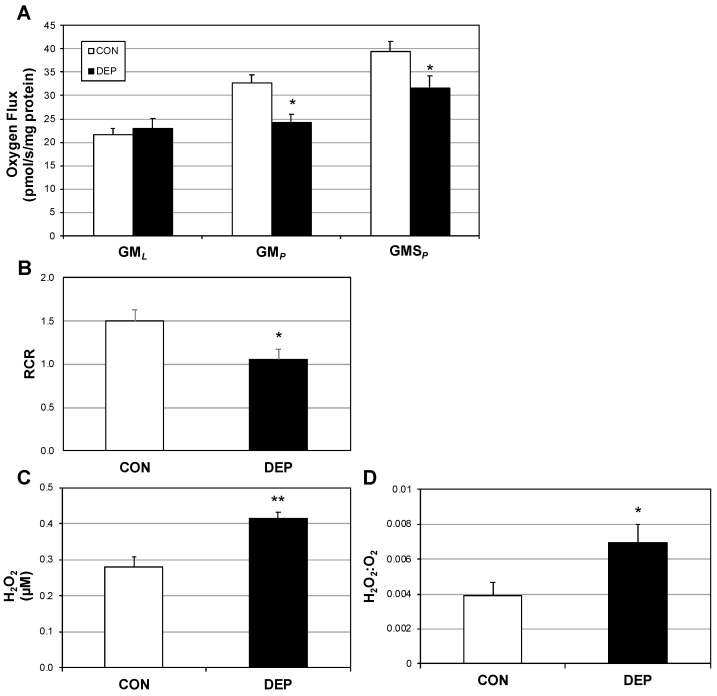
Diesel exhaust particle exposure deleteriously alters alveolar macrophage mitochondrial function in mice. Following 4 weeks of room air (CON) or diesel exposure (DEP), alveolar macrophages were obtained from mice and mitochondrial respiration (**A**) and respiratory control ratio (RCR) (**B**) were quantified. O_2_ consumption was determined according to the protocol outlined in materials and methods. GM*_L_*: glutamate (10 mM) + malate (2 mM); GM*_P_*: +ADP (2.5 mM); GMS*_P_*: +succinate (10 mM); and +antimycin A (2.5 μM; not shown) as baseline. Furthermore, H_2_O_2_ levels (**C**) and the H_2_O_2_:O_2_ ratio (**D**) were measured. *n* = 8. * *p* < 0.05, ** *p* < 0.01; CON vs. DEP.

**Figure 2 ijms-20-05598-f002:**
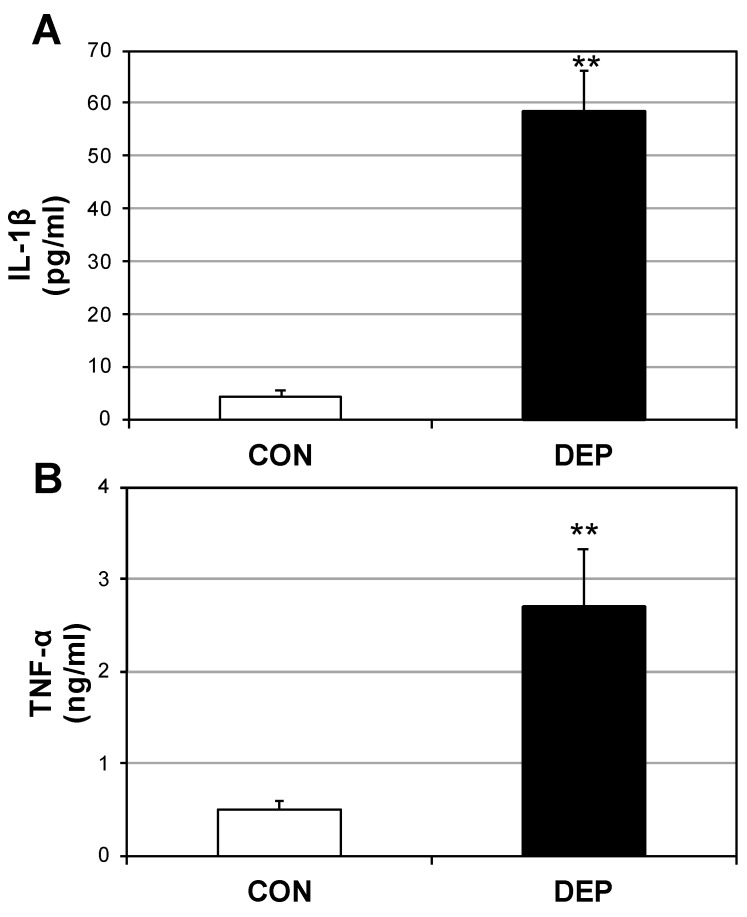
Diesel exhaust particle exposure increases circulating pro-inflammatory cytokines. Following 4 weeks of room air (CON) or diesel exposure (DEP), plasma levels of IL-1β (**A**) and TNF-α (**B**) were measured from mice. *n* = 8. ** *p* < 0.01; CON vs. DEP.

**Figure 3 ijms-20-05598-f003:**
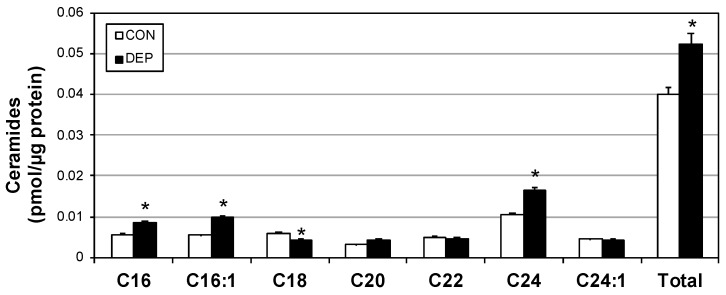
Diesel exhaust particle exposure selectively increases alveolar macrophage ceramides. Following 4 weeks of room air (CON) or diesel exposure (DEP), ceramides were quantified from mouse alveolar macrophages. *n* = 8. * *p* < 0.05; CON vs. DEP.

**Figure 4 ijms-20-05598-f004:**
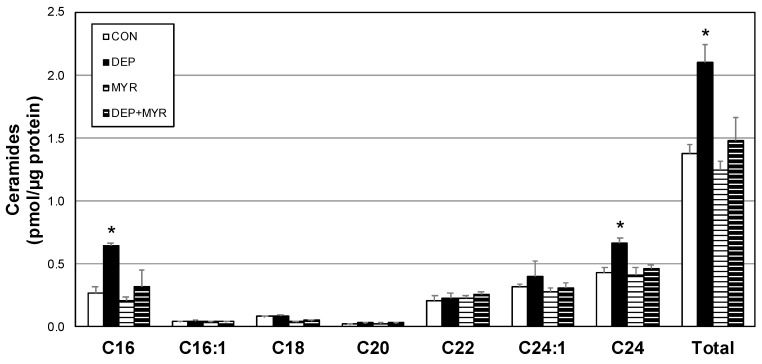
Diesel exhaust particle exposure increases ceramides in primary alveolar macrophage ceramides. Ceramides were quantified in primary alveolar macrophages after treatment with control media (CON), DEP, myriocin (MYR), or DEP + MYR for 12 h. *n* = 4. * *p* < 0.05; DEP vs. other conditions.

**Figure 5 ijms-20-05598-f005:**
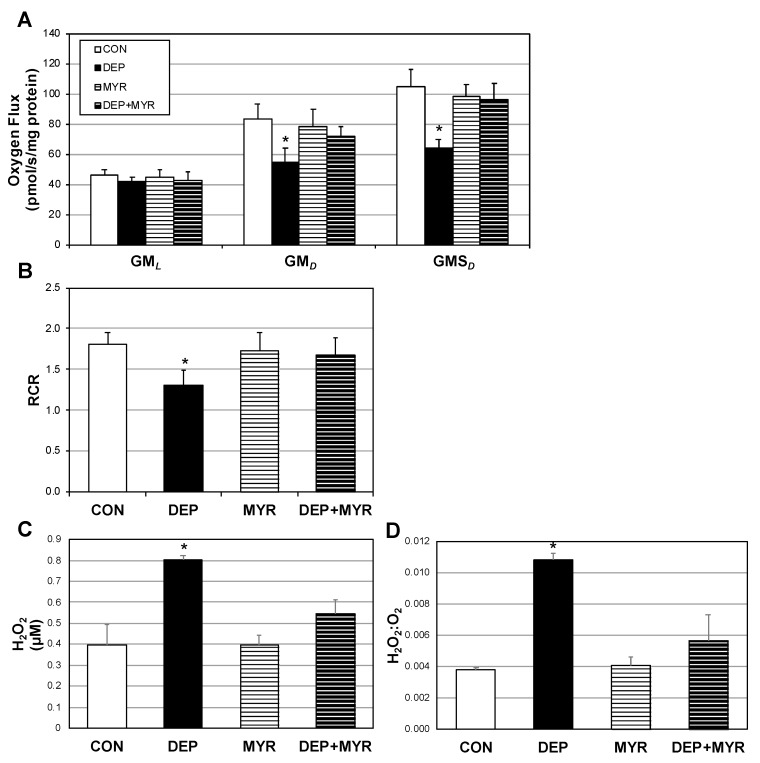
Primary alveolar macrophage mitochondria negatively respond to diesel exhaust particles (DEP). Primary alveolar macrophages were plated and treated control media (CON), DEP, myriocin (MYR), or DEP+MYR for 12 h. Following treatment, mitochondrial respiration (**A**) and respiratory control ratio (RCR) (**B**) were quantified. O_2_ consumption was determined according to the protocol outlined in materials and methods. GM*_L_*: glutamate (10 mM) + malate (2 mM); GM*_P_*: +ADP (2.5 mM); GMS*_P_*: +succinate (10 mM); and +antimycin A (2.5 μM; not shown) as baseline. H_2_O_2_ levels (**C**) and the H_2_O_2_:O_2_ ratio (**D**) were also measured *n* = 8. * *p* < 0.05; DEP vs. other conditions.

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
