# Peer review of "Diesel Exhaust Particle Exposure Compromises Alveolar Macrophage Mitochondrial Bioenergetics"

_ijms, 2019, doi:10.3390/ijms20225598_

Round 1
Reviewer 1 Report
Gibbs et al. show that daily DEP leads to alterations in mitochondrial functions and an increase of certain ceramide species in alveolar macrophages. This exposure further causes a systemic inflammation as evidenced by increased levels of pro-inflammatory cytokines as well as insulin resistance. To ascertain that the reduced RCR as a proxy of mitochondrial fitness as well as the increased H2O2 levels were cell-autonomous and driven by ceramides, the authors treated isolated alveolar macrophages with DEP alone or with DEP+MYR. Using the ceramide synthesis inhibitor was sufficient to allow the cells to retain steady-state conditions.
Albeit the manuscript contains several interesting points on the systemic and cellular changes caused by daily DEP exposure, the story is lacking a red thread. The data of alveolar macrophages is totally disconnected from the HOMA-IR results, and should be split into 2 manuscripts. This would require further experiments, e.g. is the liver/pancreas affected after DEP? What exactly is causing insulin resistance? HOMA-IR is a good first readout, but mice should be analysed in more detail by GTT/ITT (injection of glucose/insulin) at least at the endpoint. Activation of the insulin receptor after insulin injection can be assessed etc.
It remains unclear how the uptake of DEP in alveolar macrophages causes the increase of ceramides and whether this is also causal for mitochondrial dysfunction. Myriocin treatment may lead to other effects besides inhibition of ceramide synthesis (e.g. modulating the immune response). Therefore, ceramide levels of the DEP/DEP+MYR treated cells from the in vitro cell culture should be measured. Usage of CerS-knockout mice that would affect the ceramide species of interest could be another way to draw the connection of DEP-> ceramide -> mitochondrial dysfunction. In any case, the immunomodulatory functions of MYR on alveolar macrophages should be tested, as this could have an impact on mitochondrial function.
Author Response
We have analyzed and included the data of ceramide levels in primary alveolar macrophages. We have included these data as Figure 4. In sum, the results confirm the inhibition of ceramide accrual with myriocin treatment, even with DEP co-treatment.
Reviewer 2 Report
The manuscript by Gibbs et al describes the molecular effects in mice following exposure of diesel exhaust particles (DEP). Data are presented that DEP reduces respiration rates and increases hydrogen peroxide production in mitochondria from alveolar macrophages. In addition, serum ceramides and inflammatory cytokines are reported to be elevated in experimental animals exposed to DEP. Results are also outlined from cultured primary murine macrophages treated with DEP in the presence of myriocin (pharmacological inhibitor of sphingolipid biosynthesis). The authors conclude that DEP exposure may compromise macrophage mitochondrial and whole-body function through changes in macrophage ceramide metabolism. This is an interesting study, which offers a potential insight into mechanism by which air pollution might lead to cardiovascular disease. The manuscript is generally well written although there are a few points that should be considered:
1. It is important to place the current study in the context of published literature. Specifically, are the levels and duration of DEP exposure in the experimental mice typical of studies of this type? Further, how do these levels compare to human DEP exposure in urban environments?
2. Section 2.4. Could experiments be performed whereby ceramides are added to macrophage cultures and the effects of DEP subsequently studied?
3. Section 4.3. It would be useful to provide additional experimental details of the primary alveolar macrophage culture.
4. Section 4.7 Lipids. Sub-heading could be altered to ‘Ceramide Quantification’ or ‘Ceramide Analysis’.
5. Section 4.7 Lipids. Additional details on the quantitative analysis of the murine serum ceramides should be provided. The reference cited (#31) appears to relate to a short abstract and gives no information on the preparation of samples or mass spectrometric analysis.
Author Response
We have previously published work with DEP, which was used as justification for the current strategies. Importantly, we strategically used a dose of 3µg/ml, which is reported as a physiological level. We include references to our previously published work in the manuscript. We have used C2-ceramide extensively in previous manuscripts to explore the explicit effect of ceramides on mitochondrial function (and more). We conducted a tangential experiment that involved the inclusion of DEP and C2-ceramides, which revealed ceramides provided no additive effect beyond that seen with DEP alone. Because no effect was observed, we have not included these data, but included a note in the manuscript, as follows: “The addition of C2-ceramide into the culture medium provided no additive effect (data not shown).” We have included this method as follows:“To obtain primary alveolar macrophages (AMs), four WT C57Bl6 mice per experiment were sedated and exsanguinated to ensure euthanasia as outlined previously [24]. Bronchoalveolar lavage fluid (BALF) was specifically harvested through the instillation and recovery of seven 1-ml boluses of PBS with a syringe attached to a catheter for a total of 7 ml. Each 7-ml BALF sample was centrifuged at 1,000 rpm for 10 min. The cell pellet was resuspended in warm DMEM, pooled from the four mice and equal concen- trations of approximately 50,000 cells were plated and allowed to adhere overnight before exposure.”
We have updated the sub-heading and included greater details of plasma lipid extract, as follows:“For quantification of ceramides from plasma and cell lysate, 100µl of sample was resuspended in 300 μL ice-cold phosphate-buffered saline (PBS) and 1.5 mL of methanol, as described previously [27]. Samples were centrifuged and supernatant was transferred to a clean tube. Following the addition of 30 μL of 1 M KOH in methanol, samples were incubated overnight at°C. Samples were dried to 50% volume and 25 μL glacial acetic acid was added to neutralize KOH. Separation of aqueous and organic phases required addition of 300 μL LC-grade chloroform and 600 μL DDH2O followed by centrifugation for 2 min at maximum speed. The lower organic phase was transferred to a fresh vial. This separation step was repeated twice. All lipid samples were dried in a vacuum centrifuge (Eppendorf Concentrator Plus). Lipids were quantified by shotgun lipidomics using an ABI 5600+ (AB Sciex, Framingham, MA, USA). Briefly, we simultaneously identified changes in hundreds of distinct lipid species via a nonbiased approach following direct infusion of extracted lipids containing 18 mM ammonium fluoride to aid in ionization of neutral lipids and to reduce salt adducts. Data from the AB Sciex 5600+ was collected and calibrated with Analyst and PeakView Software (AB Sciex). The in-house-developed Lipid Explorer software assists with simplifying the data by identifying lipid species based on exact mass and fragmentation patterns.”

Reviewer 3 Report
More experimental results should be necessary to clarify the underlying mechanisms.
Please cite the following paper and discuss with their results.
Shiraiwa M, Selzle K, Pöschl U. Hazardous components and health effects of atmospheric aerosol particles: reactive oxygen species, soot, polycyclic aromatic compounds and allergenic proteins. Free Radic Res. 2012 Aug;46(8):927-939. doi: 10.3109/10715762.2012.663084.
Lawal AO. Diesel Exhaust Particles and the Induction of Macrophage Activation and Dysfunction. Inflammation. 2018 Feb;41(1):356-363. doi: 10.1007/s10753-017-0682-6.
H M Yang, J M Antonini, M W Barger, L Butterworth, B R Roberts, J K Ma, V Castranova, and J Y Ma Diesel exhaust particles suppress macrophage function and slow the pulmonary clearance of Listeria monocytogenes in rats. Environ Health Perspect. 2001 May; 109(5): 515–521. doi: 10.1289/ehp.01109515
Ji J, Upadhyay S, Xiong X, Malmlöf M, Sandström T, Gerde P, Palmberg L. Multi-cellular human bronchial models exposed to diesel exhaust particles: assessment of inflammation, oxidative stress and macrophage polarization. Part Fibre Toxicol. 2018 May 2;15(1):19. doi: 10.1186/s12989-018-0256-2.
As Olefsky et al. suggested that for insulin resistance, a lot of factors are included. The authors should be examined more factors to elucidate the molecular mechanisms underlying the insulin resistance. Some of KO experiments will be necessary to clarify the mechanisms of insulin resistance.
Olefsky JM, Glass CK. Macrophages, inflammation, and insulin resistance. Annu Rev Physiol. 2010;72:219-46. doi: 10.1146/annurev-physiol-021909-135846.
Abstract
Obesity induces an insulin-resistant state in adipose tissue, liver, and muscle and is a strong risk factor for the development of type 2 diabetes mellitus. Insulin resistance in the setting of obesity results from a combination of altered functions of insulin target cells and the accumulation of macrophages that secrete proinflammatory mediators. At the molecular level, insulin resistance is promoted by a transition in macrophage polarization from an alternative M2 activation state maintained by STAT6 and PPARs to a classical M1 activation state driven by NF-kappaB, AP1, and other signal-dependent transcription factors that play crucial roles in innate immunity. Strategies focused on inhibiting the inflammation/insulin resistance axis that otherwise preserve essential innate immune functions may hold promise for therapeutic intervention.
Author Response
We have included the relevant citations in the discussion. We have not included the additional citations regarding insulin resistance insofar as we have removed that aspect of this study.Reviewer 4 Report
Dr. Gibbs JL, et al demonstrated in mice that DEP (diesel exhaust particles) exposure affects on mitochondrial respiratory functions and causes insulin resistance determined by HOMA-IR. Authors have focused on an important issue, but following points should be addressed.
What is the mechanism which induced elevation of inflammatory cytokines in the blood? Relative contribution of alveolar macrophages is obscure. Does DEP per se translocate to the circulating blood and induce activation of inflammatory cells? How activation of alveolar macrophages by DEP resulted in systemic inflammatory milieu?
The link between alveolar macrophages and insulin resistance is also missing. Are there any changes in adipose tissue inflammation or pancreas after DEP exposure?
In Figure 3, the change of ceramides level seems to be modest. This point raises a concern whether ceramides actually play critical roles in the induction of mitochondrial functional changes and consequent metabolic changes. How inhibition of ceramides biogenesis canceled the effects of DEP on mitochondrial respiratory function?
How about other indexes related to diabetes including glucose tolerance test and insulin tolerance test?
Author Response
Our previous work with DEP implicates RAGE is a necessary intermediate in DEP-induced inflammation. We have cited this work in the manuscript. Yes, DEP moves into the bloodstream, making it a challenge for us to directly implicate lung, rather than systemic, macrophages in promoting systemic inflammation. To strengthen to focus of this project, we have removed any discussion and data regarding insulin resistance. The in vivo plasma ceramides are indeed modestly altered between the groups. We implicate ceramides due to the protective effect of treatment with myriocin, which inhibits ceramide biosynthesis and accrual. If the changes in ceramides were irrelevant, however slight they may be, we would have expected myriocin co-treatment to yield no protective effect.Round 2
Reviewer 1 Report
The authors show now a compelling story with the focus on alveolar macrophages.
Minor points remain to be addressed:
The introduction should have a paragraph on alveolar macrophages and what is know so far about the impact of DEP on these cells. The authors should elaborate on the usage of C2-ceramides (lines 103-104). E.g. what is the objective of doing this experiment? Even if they are negative results, they could be shown in a supplemental figure. Also, this is not described in the methods section.Author Response
We have included the requested paragraph in the Introduction, which reads as follows:“Inhaled diesel exhaust elicits potent effects on the body, rapidly activating myriad inflammatory processes as the diesel particles (DEP) affect varying levels of the respiratory tree [6]. Unsurprisingly, consistent DEP inhalation is implicated with several cardiorespiratory disorders and mortality [7], with both epidemiological [8] and molecular [7] evidence indicating a primary role of the lung macrophage. At least some of the pathology associated with noxious stimuli exposure is a result of macrophage-induced cytokine release, which has been a focus of earlier work from our lab [4, 9, 10].
We have included the use of C2-ceramide in the methods and elaborated the results to describe our negative finding. We have not included the results as a supplement as we feel it is not warranted and adds little value. If the editor insists, we will include data as a supplemental figure.
Reviewer 3 Report
Title: mitochondrial physiology is too wide. Please change to mitochondrial respiration.
Abstract:Delete “Once again, not only was mitochondrial respiration diminished following DEP exposure, but H2O2 production increased significantly.”
From where H2O2 emit? And how?
The authors focus mitochondrial respiration and emission of H2O2. What the authors think the relevance of mitochondrial respiration and emission of H2O2. These should be discussed.
The authors should examine MnSOD, CuZnSOD and ECSOD activity under DEP exposure.
Author Response
We have replaced the word “physiology” with “bioenergetics”, which we feel more fully encompasses both the respiration and ROS data. We altered the sentence in the Abstract: “We observed similar changes as seen in pulmonary macrophages, namely diminished mitochondrial respiration, but increased H2O2” Our data do not allow us to directly identify the source of H2O2. We have included the following paragraph in the discussion:“Alveolar macrophages exist in an oxygen-rich milieu. We submit that the H2O2:O2 ratio is a valuable outcome in identifying the use of oxygen in alveolar macrophages; namely, whether oxygen use in the mitochondria is used for the production of energetic (e.g., ATP) or signaling (e.g., H2O2) molecules. Importantly, oxidative stress molecules (i.e., O2-, H2O2, etc.) induce the expression of inflammatory signals in macrophages [27]. Future efforts will reveal the degree to which reactive oxygen species are necessary in DEP-induced macrophage cytokine production.“
We hope to follow up this study with greater scrutiny of the mitochondria in alveolar macrophages, including the role of mitochondrial morphology/dynamics, and anti-oxidant enzymes (i.e., SOD) and compounds (i.e., N-acety cystine).
Reviewer 4 Report
Authors have responded to the reviewers’ comments point by point and improved overall quality of the manuscript.
Author Response
Thank you for the positive and encouraging review.Round 3
Reviewer 3 Report
The manuscript became much better. It could be acceptable now.